# Effects of Heat Stress on Follicular Physiology in Dairy Cows

**DOI:** 10.3390/ani11123406

**Published:** 2021-11-29

**Authors:** Fabio De Rensis, Roberta Saleri, Irina Garcia-Ispierto, Rex Scaramuzzi, Fernando López-Gatius

**Affiliations:** 1Department of Veterinary Sciences, University of Parma, Strada del Taglio 10, 12, 43121 Parma, Italy; fabio.derensis@unipr.it; 2Department of Animal Science, University of Lleida, 25198 Lleida, Spain; irina.garcia@udl.cat; 3Agrotecnio Centre, 25198 Lleida, Spain; lopezgatiusf@gmail.com; 4Royal Veterinary College, London NW1 0TU, UK; rex.scaramuzzi@orange.fr; 5Institute of Agriculture, University of Western Australia, Perth 6009, Australia; 6Transfer in Bovine Reproduction SLu, 22300 Barbastro, Spain

**Keywords:** follicular cooling, graafian follicle, oocyte competence, ovulation failure, temperature differential

## Abstract

**Simple Summary:**

Environmentally induced hyperthermia, also called heat stress (HS), compromises reproductive physiology in mammals. The number of oocytes is fixed after birth and they are stored in the ovary in a quiescent state (at the stage of the first meiotic prophase) in primordial follicles. There is evidence that HS alters the oocyte quality, the dynamics of follicular growth and ovulation. The dairy cow, submitted to the metabolic stress of high milk production, is a good model for studying the effects of HS on ovarian function. The aim of this review is to describe the influence of HS during the stages of follicular development in dairy cattle, from the activation of primordial follicles to ovulation. Some clinical aspects are also considered.

**Abstract:**

Follicular organization starts during mid-to-late fetal life with the formation of primordial follicles. The bilateral interplay between the oocyte and adjoining somatic cells during follicular growth and ovulation may be sensitive to heat stress (HS). Mechanisms giving rise to pre-ovulatory temperature gradients across reproductive tissues are mostly regulated by the pre-ovulatory follicle, and because the cooling of the gonads and genital tract depends on a counter-current transfer system of heat, HS may be considered a major factor impairing ovulation, fertilization and early embryo development. There is evidence of a long-lasting influence of HS on oogenesis and final follicular maturation. Follicular stages that are susceptible to HS have not been precisely determined. Therefore, the aim of this review was to describe the influence of HS during the staged follicular development in dairy cattle, from the activation of primordial follicles to ovulation. Some clinical prospects are also considered.

## 1. Introduction

Environmentally induced hyperthermia (heat stress (HS)) compromises reproductive physiology in mammals (reviewed in [1]). In lactating dairy cows, the assumed upper critical temperature under shade is about 25–28 °C, with a maximum temperature-humidity index (THI) of 78.2 [2,3]. Above these given values, the rectal temperature rises, and the conception rate can decline by an average of 20 to 30% [4,5]. Some of the reasons that explain this decline include a greater load of metabolic heat to dissipate in lactating than in non-lactating cows [6], compromised follicular function and a higher incidence in early embryonic loss [7,8]. Consequently, a better understanding of the physiological mechanisms that can adapt the reproductive system under HS conditions is currently a research priority. The concept of an environmental ‘heat wave’, based on a sequence of at least five consecutive days during which air temperatures and relative humidity are higher than usual, has been recently developed to help evaluate the magnitude of the influence of HS on fertility [9,10,11].

The neuroendocrine mechanisms by which HS can compromise the reproductive efficiency of the dairy cow has been extensively reviewed [12,13,14]. In summary, HS compromises hypothalamic GnRH secretion, and thus, the circulating concentrations of LH and FSH are reduced [12,13]. It follows that this reduced cascade of neuroendocrine and endocrine events leads to impaired selection and development of ovulatory follicles, ovulation and the development of a functional corpus luteum. Where the deleterious effects of HS on oocytes and spermatozoa have been compared, it seems that the main cause of fertilization failure is oocyte quality, rather than that of spermatozoa [14,15]. In addition, there is also evidence that HS alters the dynamics of follicular growth and ovulation [1,16]. As the follicular stages that are susceptible to thermal stress have not been precisely determined, the aim of this review was to describe the influence of HS during staged follicular development in dairy cattle, from the activation of primordial follicles through to ovulation. Some clinical prospects are also considered.

## 2. Follicular Development under Conditions of Heat Stress

In mammals, the number of oocytes is fixed after birth and they are stored as a quiescent state (the first stage of meiotic prophase) in primordial follicles. Following the normal onset of puberty, the regulation of the activation process of the primordial-to-primary follicle transition to support an ovulatory cycle is not fully understood [17]. Possible influences of HS on the regulatory mechanisms of maintaining the population of primordial follicles or their regulated activation are largely unknown.

### 2.1. Primary and Preantral Follicles

There is no direct evidence of an influence of HS on primary follicles. However, the fact that restoration of normal fertility is only achieved 40–60 days after a period of HS [18] suggests a negative influence of HS on the very early stages of follicle development. In effect, cattle fertility is compromised not only in summer, but also in the autumn and the early winter, when the temperature is temperate [19,20]. In a recent study, all stages of growing preantral follicles, including primary and secondary follicles, were susceptible to negative effects of HS in vitro [21]. Therefore, a many-sided negative influence of HS on preantral follicles should be expected; the estimated duration of follicular growth from the primary to pre-ovulatory stage is around 60 days [22]. The delicate processes of follicular development are probably sensitive to HS, which include oocyte development during the primary to secondary follicle transition. At this time, zona pellucida material is deposited around the oocyte, cortical granules are produced within the oocyte cytoplasm and follicles appear to become responsive to gonadotropins [22,23]. It is at this point of development that the bilateral interplay between the oocyte and surrounding pre-granulosa cells becomes decisive until the follicle is fully grown or falls into atresia [22,23,24]. The oocyte is the first cell within the follicle to be affected by apoptosis, but if it overcomes this risk, it will grow actively up to the preantral stage [23]. The growth phase of an oocyte is critical in terms of its competence not only for fertilization and embryogenesis but also for final follicle maturation and ovulation. Oocytes seem to play key roles in controlling granulosa cell development and function from the time of follicular organization through to ovulation [24,25]. This bilateral interplay between the oocyte and adjoining somatic cells from the time of follicular organization to the final process of follicle maturation and ovulation is probably sensitive to HS (See Figure 1 for a summary).

### 2.2. Maturation of Oocytes and Antral Follicles

Studies on follicular growth rates [26,27] suggest that a period of two to three estrous cycles (i.e., four to six follicular waves) are required for recovery from summer heat damage to the follicle population and the appearance of competent, mature oocytes in the subsequent autumn. This seems to be associated with a long-lasting effect of HS on follicular steroidogenesis associated with a lower secretion of androstenedione by the theca cells and compromised aromatase activity of the granulosa cells of the dominant follicle [7,20]. In essence, the early antral follicles of approximately 0.5 to 1.0 mm in diameter [19] or those in the early growth phase of the follicular wave [20,28,29] are sensitive to HS. The size of the dominant follicle is reduced in cows under acute HS [30,31,32,33,34], causing an attenuation of dominance and an increase in the number of large-sized non-dominant follicles [19,28,32,35]. Furthermore, this compromised follicular physiology is inferred by the fact that bovine oocytes collected during the summer experience a greater incidence of disruption in maturation [36,37,38] expressed as a reduced proportion of embryos that develop to the blastocyst stage [36,37,38]. Although the sensitive systems of temperature regulation in vivo are difficult to mimic in vitro, the culture of individual Graafian follicles [39,40] under different levels of HS might reveal any perturbation in oocyte maturation. Biochemical changes in follicular fluid of the periovulatory follicle in lactating dairy cows exposed to HS have also been described [41,42]. From a clinical perspective, removal of impaired follicles from previously heat-stressed cows may led to an earlier emergence of healthy follicles and high-quality oocytes [30].

### 2.3. Pre-Ovulatory Follicle and Ovulation

As expected, following the reduced size of the dominant HS follicle will influence the diameter of the pre-ovulatory follicle. The size of the pre-ovulatory follicle is reduced with a high environmental THI value [43]. In fact, the diameter of the pre-ovulatory follicle is reduced by an estimated 0.1 mm for each additional point on the THI value on the day of estrus [44]. A low level of estradiol production may explain the high incidence of ovulation failure, both in synchronized and spontaneous estrus, in cows under HS conditions [45,46]. Estradiol concentrations rise in pre-ovulatory follicles until a required threshold, which determines both estrus behavior and the pre-ovulatory LH surge [47]. Furthermore, exposure of lactating dairy cows to acute pre-ovulatory HS has been associated with a reduction of granulosa cell-specific gene expression in follicles of ovulatory size [48]. Although the dominant follicle acquires the capacity to ovulate when it reaches a diameter of about 10 mm [49], a smaller pre-ovulatory follicle in monovular cows (10–15 mm) has also been associated with ovulation failure [50,51] (see Figure 2 for a summary). Mechanisms related to ovulation failure may be better understood by studying the process of pre-ovulatory cooling of the reproductive system.

## 3. Temperature Gradients in the Reproductive System

Cooling during the peri-ovulatory period of segments of the genital tract is related to oocyte maturation and fertilization [52,53], whereas reduced cooling of the genital tract may compromise fertilization and early embryonic development [1,30]. In essence, the temperature increases progressively from the vagina to the uterine horns [54] and to a cooler microenvironment in the caudal isthmus of the oviduct [52]. The caudal isthmus is recognized as the reservoir for fertilizing spermatozoa and a place where sperm motility is reduced [55,56,57]. Thus, ejaculated spermatozoa are exposed to an increasing temperature from the cranial vagina to the oviduct sperm reservoir. Sperm migration occurs from the cooler caudal isthmus to the warmer site of fertilization, close to the isthmus-ampullary junction. Conversely, oocytes are shed from a cooler follicle and are transported to a warmer ampulla on their path to the site of fertilization [52,53]. The thermal interplay between ovary, oviduct and uterus is facilitated by counter-current exchange systems of heat exchange among veins, interstitial fluid, lymph and arteries [58,59,60]. The enormous amount of periovulatory endometrial and cervical mucus secretion and flow [61] may also favor a temperature decrease from the uterine horns to the vagina [16]. Temperature changes in the oviducts, uterus and vagina are mostly regulated by the ovaries.

### Periovulatory Changes in Ovarian Temperature

Intra-follicular cooling during the pre-ovulatory period acts to regulate mammalian ovulation [25]. A large body of evidence indicates that pre-ovulatory follicles may be over 1 °C cooler than ovarian stromal tissues, with both ovarian components being cooler than in neighboring tissues and rectal temperatures in sheep [62], rabbits [63], humans [64], pigs [65,66] and cows [46,67,68,69]. Such cooling correlates positively with the potential for pregnancy in humans [64] and cows [69]. In fact, a functional ovary requires a lower temperature than neighboring tissues [70]. However, environmental HS along with high milk production jeopardizes the ovulatory process [1,30]. Studies of follicular cooling under HS conditions in lactating dairy cows show that when follicles of pre-ovulatory size were cooler than the deep rectal temperature they ovulated, whereas follicles of a similar size but not showing a temperature differential did not ovulate [46,68,69]. The process of follicular cooling is very sensitive to HS (Figure 2). Cows exposed to solar radiation for a period of less than 45 min just after sunrise (dawn) failed to ovulate, whereas most of the cows in the shade ovulated [46]. In this study, fans and water sprinklers were not yet activated during the time of the temperature measurements (Figure 3). A damaged oocyte is probably responsible for follicular cooling failure and subsequent ovulation failure. The fact that an oocyte promotes cooling of a pre-ovulatory follicle has been recently hypothesized [25].

Data derived from a more recent study showed that the temperature differential between the caudal cervical canal and rectum at AI may be a predictive factor for pregnancy [71]. A low cervical temperature at AI was associated with a greater likelihood of pregnancy, whereas vaginal temperatures and vagina-cervix and vagina-rectum temperature differentials at AI and 7 days post-AI could not be related to the likelihood of ovulation or pregnancy [71]. Given the difficulties in measuring follicular temperature under field conditions, cervix-rectum temperature differentials could be used in routine veterinary practice in dairy herds, at least as an additional tool to confirm estrus ready for service. Temperature differential of the cervical canal was not recorded in cows suffering ovulation failure. Thus, in cows not showing low cervical temperature at AI, an elevated dose of GnRH at insemination should bring ovulation forward and increase their fertility [72].

## 4. Clinical Implications

The control of reproductive programs in dairy cattle should include measures of the incidence of non-cyclicity at finishing the waiting period, ovulation failure in cows showing estrus and the distribution of embryos in multiple pregnancies. These parameters could be used to evaluate influence of HS on follicular physiology and subsequent fertility. For example, from extensive studies in which results of a warm period were compared with those of a cool period, during the warm period of the year (in parentheses, percentages of the cool versus the warm period are shown), non-cyclicity and the ovulation failure rate were increased, *n* = 12,711 cows starting the AI period; 6.5–26.4% [73] and *n* = 1917 cows showing estrus signs; 3.4–12.4% [45], respectively, and the incidence of unilateral multiple pregnancies (at least two embryos in the right or left uterine horn) increased compared to bilateral multiple pregnancies (at least one embryo in each uterine horn) (*n* = 1130 cows carrying multiple pregnancies; 51.7–58.4%) [74]. In the latter study, the unilateral/bilateral multiple pregnancy ratio varied from 1.07 (346/323) during the cool period to 1.4 (269/192) during the warm period [74]. In other words, a ratio between unilateral and bilateral multiple pregnancies close to one may be a good indicator of cow well-being and absence of HS in dairy herds [75].

## 5. Conclusions

Probably, oocyte and follicular development are not only susceptible to HS, but also to any other forms of stress. However, global warming is already impairing physiological functions in mammals [1] and HS is presented here as a model of stress in the study of ovarian follicular physiology. As cooling of the gonads and genital tract depends on a counter-current transfer system of heat, HS may be considered a major factor impairing ovulation, fertilization and early embryo development. Although knowledge on how high environmental temperatures affect reproduction is increasing rapidly, a great deal of practical improvement still needs to be made to reduce the negative effects of HS in high-producing dairy herds [5,76,77]. The long-lasting effect of HS on oocyte development and competence suggests that environment should be controlled throughout the productive lifespan of these cows, starting before they are even born. In effect, HS during gestation also has a negative effect on the subsequent survival and reproductive performance of newborns [77,78,79]. Mammalian oocytes and follicle development starts during mid-to-late fetal life with the formation of primordial follicles [80]. The stage of first meiotic prophase when the development of the oocyte is arrested occurs during fetal life does not resume until shortly before ovulation [81]. This long period of interrupted meiosis may well be sensitive to HS, impairing follicular growth. However, definitive proof of this assertion is not yet available.

## Figures and Tables

**Figure 1 animals-11-03406-f001:**
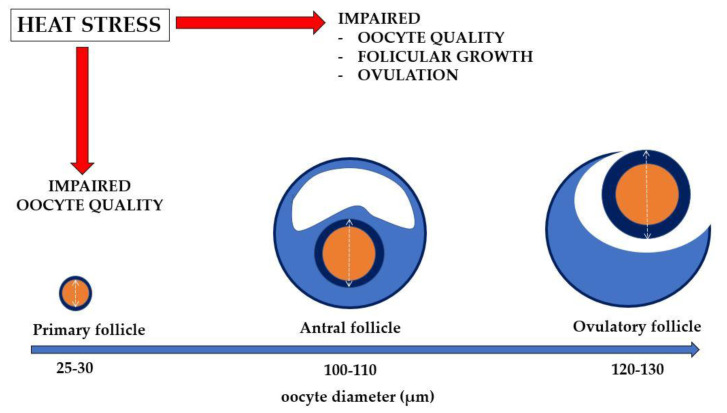
Schematic-model of oocyte and follicular development. Oocyte diameter based on data from Lussier et al. [26] and Hulshof et al. [27].

**Figure 2 animals-11-03406-f002:**
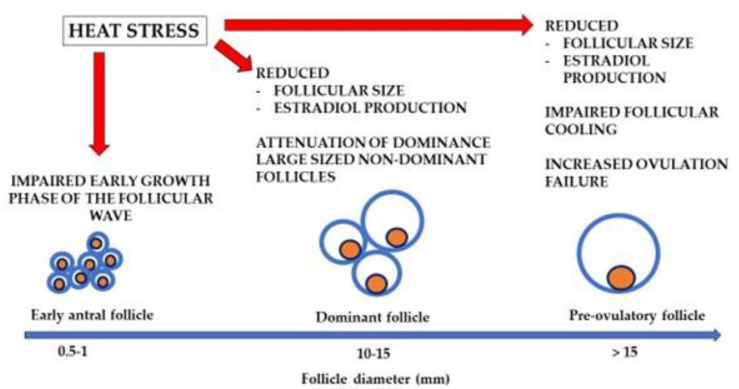
Schematic-model of follicular growth. Follicle diameter based on data from Lusier et al. [26] and Hulshof et al. [27].

**Figure 3 animals-11-03406-f003:**
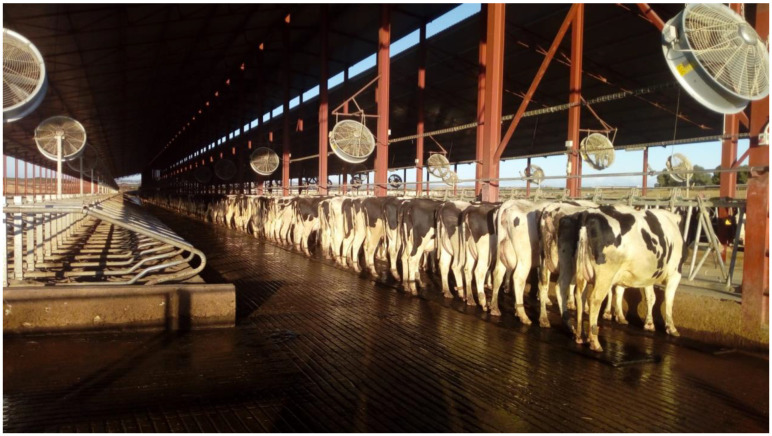
Cows in the shade or solar radiation areas [46] [photo by F. Lòpez-Gatius].

## Data Availability

Not applicable.

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
