# Peer review of "Effects of Heat Stress on Follicular Physiology in Dairy Cows"

_animals, 2021, doi:10.3390/ani11123406_

Round 1

Reviewer 1 Report

Overall comments: The manuscript “Effects of Heat Stress on Follicular Physiology in Dairy Cows” brings information from recent literature focusing on heat stress in the follicular environment. It approaches one of the main problems on bovine reproduction, the exposure to high temperatures leading to heat stress. In fact, the heat stress combines several variables, such as temperature, humidity, wind, etc… These factors influence directly and indirectly the beef and dairy industry. Indeed, the impact of heat stress on follicular and oocyte quality may last longer, and for this reason may have a major role in this topic. It is well written and takes new information regarding to the topic. I believe the quality of the figures does not match the quality of the text. For this reason I would like figures to be improved.

Originality: The review brings together the impact of heat stress on follicular development. It is well structured and leads the reader through the physiological events. I would suggest to include a more detailed subsection on oocyte quality.

Appropriate conclusions: Conclusions are appropriate. Nevertheless, I would recommend the authors to include a sentence or two of suggestions to minimize the impacts of heat stress specific on follicular development (review topic).

Decision: Accept.

Author Response

[Reviewer(s)' Comments]

Reviewer 1

Overall comments: The manuscript “Effects of Heat Stress on Follicular Physiology in Dairy Cows” brings information from recent literature focusing on heat stress in the follicular environment. It approaches one of the main problems on bovine reproduction, the exposure to high temperatures leading to heat stress. In fact, the heat stress combines several variables, such as temperature, humidity, wind, etc… These factors influence directly and indirectly the beef and dairy industry. Indeed, the impact of heat stress on follicular and oocyte quality may last longer, and for this reason may have a major role in this topic. It is well written and takes new information regarding to the topic. I believe the quality of the figures does not match the quality of the text. For this reason, I would like figures to be improved.

(Au): Thank you very much for your consideration. Figures 1 and 2 have been changed and quality improved.

Originality: The review brings together the impact of heat stress on follicular development. It is well structured and leads the reader through the physiological events. I would suggest to include a more detailed subsection on oocyte quality.

(Au) Regarding oocyte quality, lack of knowledge on the effects of HS on oocyte/follicle quality are noted in different points (Lines 71–76) and in the last sentence of the conclusions section. We think that a more detailed description on oocyte quality is out of this review.

Appropriate conclusions: Conclusions are appropriate. Nevertheless, I would recommend the authors to include a sentence or two of suggestions to minimize the impacts of heat stress specific on follicular development (review topic).

(Au) Some points have been added at the end of different sections: Lines 123–125; 210–212 and a new subsection on clinical implications has been created, as suggested by the Reviewer 3, where the former second paragraph of the conclusions has been translated to this place.

Thank you very much for your perceptive and constructive criticism.

Reviewer 2 Report

The review is focused on the effect of heat stress on follicular development in dairy cows, from the early stages of development until ovulation. The paper is very interesting, well written, and covers many aspects 
of this matter not forgetting references to the clinical aspects, that is a strength point of the paper.
I have only one request and some very minor revisions.
I have read some papers reporting the effect of heat stress on follicular fluid composition; e.g
- Biochemical changes in the follicular fluid of the dominant follicle of high producing dairy cows exposed to heat stress early post-partum. Shehab-El-Deen et al, Anim Reprod Sci. 2010 Feb;117(3-
4):189-200. doi: 10.1016/j.anireprosci.2009.04.013. 
- Heat-induced hyperthermia impacts the follicular fluid proteome of the periovulatory follicle in lactating dairy cows. Ripoli et al, PLoS One. 2019 Dec 30;14(12):e0227095. doi: 
10.1371/journal.pone.0227095
I think it would be interesting to add some discussion regarding this point, as HS influences not only the cells (oocytes) but also the composition of the fluids in which they are placed.
Some minor check:
Line 83: change a with at
Line 89: seem TO play
Line 136: check grammarly the sentence The size and estradiol production dominant follicle
Line 137-139: check also the sentence The diameter of the preovulatory follicle is reduced by an estimated 0.1 mm for each additional point on the THI value on the day of estrus has been estimated in pre-ovulatory follicles

Author Response

[Reviewer(s)' Comments]

Reviewer 2

The review is focused on the effect of heat stress on follicular development in dairy cows, from the early stages of development until ovulation. The paper is very interesting, well written, and covers many aspects of this matter not forgetting references to the clinical aspects, that is a strength point of the paper.

(Au): Thank you very much for your consideration.

I have only one request and some very minor revisions.

I have read some papers reporting the effect of heat stress on follicular fluid composition; e.g

- Biochemical changes in the follicular fluid of the dominant follicle of high producing dairy cows exposed to heat stress early post-partum. Shehab-El-Deen et al, Anim Reprod Sci. 2010 Feb;117(3-4):189-200. doi: 10.1016/j.anireprosci.2009.04.013.

- Heat-induced hyperthermia impacts the follicular fluid proteome of the periovulatory follicle in lactating dairy cows. Ripoli et al, PLoS One. 2019 Dec 30;14(12):e0227095. doi: 10.1371/journal.pone.0227095

I think it would be interesting to add some discussion regarding this point, as HS influences not only the cells (oocytes) but also the composition of the fluids in which they are placed.

(Au) Thank you very much for your comment and references. This point has been added with the suggested references (Lines 121-123).

Some minor check:

Line 83: change a with at

Line 89: seem TO play

Line 136: check grammarly the sentence The size and estradiol production dominant follicle

Line 137-139: check also the sentence The diameter of the preovulatory follicle is reduced by an estimated 0.1 mm for each additional point on the THI value on the day of estrus has been estimated in pre-ovulatory follicles

(Au) Changes have been performed.

Thank you very much for your perceptive and constructive criticism.

Reviewer 3 Report

The authors reviewed the effects of Heat Stress on Follicular Physiology in Dairy Cows. 

The manuscript is well organized and clear to follow. However many of the citations rely on other reviews, rather than in the original works.
In my opinion, the authors should go further in two fields:

1 -the hormonal and molecular events that affect follicular physiology under HS;

2 - Nutritional and other management strategies to overcome the impact of HS in follicular physiology.

Other comments and suggestions are made directly in the attached pdf file. 

Author Response

Reviewer(s)' Comments]

Reviewer 3

The authors reviewed the effects of Heat Stress on Follicular Physiology in Dairy Cows.

The manuscript is well organized and clear to follow. However, many of the citations rely on other reviews, rather than in the original works.

(Au): Thank you very much for your consideration. Seven reviews have been removed (old references 1, 2, 6, 11, 31, 58 and 59) and three new articles added. We used original works with results particularly relevant for the manuscript. We think that a more extensive revision on HS influencing reproductive physiology could be out of the scope of this review.

In my opinion, the authors should go further in two fields:

1 -the hormonal and molecular events that affect follicular physiology under HS;

2 - Nutritional and other management strategies to overcome the impact of HS in follicular physiology.

(Au) Lack of knowledge on the effects of HS on follicle physiology are noted in different points (Lines 71–76) and in the last sentence of the conclusions section. We tried to describe the sensitivity of follicular growth to HS in a balanced form for all follicular stages.

Lines 123–125; 210–212, and the new subsection 4 on clinical implications suggested in the PDF file, provide some strategies to mitigate/evaluate the influence of HS on follicular physiology.

Other comments and suggestions are made directly in the attached pdf file.

(Au) All points have been taken into account and revised accordingly.

Thank you very much for your perceptive and constructive criticism.

Round 2

Reviewer 3 Report

Now everything is in place and could be published.